# Degradation of Triclosan in the Water Environment by Microorganisms: A Review

**DOI:** 10.3390/microorganisms10091713

**Published:** 2022-08-25

**Authors:** Yiran Yin, Hao Wu, Zhenghai Jiang, Jingwei Jiang, Zhenmei Lu

**Affiliations:** 1MOE Laboratory of Biosystem Homeostasis and Protection, College of Life Sciences, Zhejiang University, Hangzhou 310058, China; 2Zhejiang Haihe Environmental Technology Co., Ltd., Jinhua 321012, China

**Keywords:** triclosan, water environment, biodegradation, molecular mechanisms

## Abstract

Triclosan (TCS), a kind of pharmaceuticals and personal care products (PPCPs), is widely used and has had a large production over years. It is an emerging pollutant in the water environment that has attracted global attention due to its toxic effects on organisms and aquatic ecosystems, and its concentrations in the water environment are expected to increase since the COVID-19 pandemic outbreak. Some researchers found that microbial degradation of TCS is an environmentally sustainable technique that results in the mineralization of large amounts of organic pollutants without toxic by-products. In this review, we focus on the fate of TCS in the water environment, the diversity of TCS-degrading microorganisms, biodegradation pathways and molecular mechanisms, in order to provide a reference for the efficient degradation of TCS and other PPCPs by microorganisms.

## 1. Introduction

Triclosan (5-chloro-2-(2’,4’-dichlorophenoxy) phenol, TCS) is an antibacterial agent that has been widely used in detergents, cosmetics, soaps, toothpaste and other pharmaceuticals and personal care products (PPCPs) since it was first marketed in 1957 [1,2]. TCS is a synthetic chlorinated aromatic compound, its basic physical and chemical properties are shown in Appendix A. In the late 1930s and early 1940s, it was found that the substitution on aromatic rings of hydrogen atoms with chlorine can yield a kind of powerful biocides, including antimicrobials [3]. TCS is a kind of broad-spectrum antimicrobial, and it is widely found in surface water, soil and sediment [4,5]. Although the U.S. Food and Drug Administration issued the final rule on the safety and effectiveness of consumer preservatives in 2016, requiring the prohibition of the use of antimicrobials such as TCS in PPCPs such as soap, it still retains a large market demand [6].

TCS is a small molecular compound with controversial roles [7]. The bacteriostatic action of TCS preventing microbial growth is due to the inhibition of the enoyl–acyl carrier reductase, which is an enzyme involved in fatty acid synthesis [7,8]. However, the greatest concern for the potential health effects of TCS is related to its endocrine-disrupting activity [9]. TCS can also harm digestive, endocrine and reproductive development systems of aquatic organisms and human bodies [10,11,12,13].

In recent years, with the widespread use of TCS, the level of TCS in the water environment has been increasing, especially since the COVID-19 pandemic [14]. Wastewater discharge is the main source of TCS in the water environment. As one of the most concerned emerging pollutants in the world, TCS has also received extensive attention in its remediation research. In nutrient-poor aquatic or terrestrial ecosystems, TCS’s natural decay rate is very slow [15]. At present, some physical and chemical remediation methods such as photolysis, adsorption and oxidation have been applied for the removal of TCS in the environment. However, it is difficult to remove TCS in the actual treatment process due to its easy production of intermediates and high maintenance cost [15,16,17]. Studies have shown that TCS could be degraded by the ammonia-oxidizing bacteria (AOB) and heterotrophic bacteria in the water environment, and using microbial metabolism to degrade TCS in wastewater can lead to complete mineralization of TCS without producing toxic intermediate metabolites [1,18]. At present, with the great progress of isolation and culture technology, more and more TCS-degrading microorganisms and TCS-degrading microbial consortia have been obtained in the laboratory, and their degrading characteristics have been extensively studied.

We searched literatures with the theme of “triclosan degradation”, “triclosan biodegradation”, and “triclosan degrading bacteria” in the Web of Science (WOS), and retrieved 1630 articles and 79 reviews, 375 articles and 24 reviews, and 92 articles and 2 reviews, respectively. We counted the number of literatures published on different topics from 2000–2021 (Appendix A). When writing this review, we searched 110 studies in WOS, including 45 literatures related to TCS biodegradation and TCS-degrading bacteria, including journals related to environmental microbiology such as Water Research, Chemosphere, Applied Microbiology and Biotechnology (Appendix A). We mapped the top 10 frequently used keywords in these 45 references and their co-occurrence network, and found that the current research related to this topic mainly focuses on TCS degradation, biodegradation and wastewater (Appendix A).The reviews related to TCS in the recent five years mainly focus on the harm of TCS to organisms [19], resistance of microorganisms to TCS [20,21,22,23] and degradation of TCS [15,17,24,25]. TCS-degrading methods mainly include adsorption [17], oxidation [26], and biodegradation [15,24], etc. The latest review about TCS biodegradation was published in 2019 [15], and after 2019, new TCS-degrading microorganisms and degradation pathways were still reported. But details of biodegradation processes, molecular mechanisms and future research trends have not been reviewed.

Therefore, in this review, we focus on the research progress of TCS biodegradation by single microorganisms and microbial consortia, and discuss the research methods and future trends of TCS biodegradation, so as to understand the microbial-mediated TCS-biodegrading mechanisms in the water environment and its applications in remediation.

## 2. Occurrence of TCS in the Water Environment

TCS in the water environment comes from the use of the hospital disinfectants, factory production of PPCPs wastewater and the use of PPCPs by humans. Wastewater containing TCS is discharged into Wastewater Treatment Plants (WWTPs), part of which is degraded by microorganisms, another is discharged into natural water with the effluent of WWTPs, and the other is discharged into the soil in the form of excess activated sludge. TCS in the soil enters the surface water and groundwater through the action of rainwater, and the surface water and groundwater are used as reference water sources [27]. The TCS enters the water purification plant, and the treated water is used as daily water [27]. This achieves the cycle of TCS in the water environment (Figure 1A).

In the water environment, TCS will be transformed by photolysis into toxic substances such as 2,3-dichlorodibenzo-*p*-dioxin, dibenzodioxin and phenol intermediates, which are harmful to the aquatic organisms by bioaccumulation [28,29,30,31] (Figure 1B). Human beings ingest TCS through direct skin contact or consumption of TCS-containing products (e.g., fish that accumulate TCS), breathing TCS-containing air, and drinking TCS-containing water (Figure 1C).

Globally, it is estimated that 1500 tons of TCS produced annually, with approximately 132 million liters of TCS-containing products used in the U.S. alone [19]. The TCS productivity was significantly increased due to the high demand for disinfection since the outbreak of COVID-19 pandemic [25,34,35]. About 80% of TCS comes from cosmetics, household cleaning products and various PPCPs [32,36]. Domestic wastewater generated by humans through the use of TCS-containing PPCPs is considered to be a major source of TCS in wastewater [1,37].

After the wastewater-containing TCS is treated by WWTPs, part of TCS is adsorbed by the sludge, and the other part is degraded by microorganisms. The concentrations of TCS in the influent, effluent and sludge of some WWTPs are shown in Table 1.

The way TCS enters the natural aquatic environment is mainly through wastewater discharge, so it is necessary to study the removal of TCS in WWTPs. As shown in Table 1, the presence of TCS was detected in both the influent and effluent of WWTPs. TCS was also detected in the sludge (Table 1). Heidler et al. [43] studied the removal effect of TCS by traditional WWTPs using a mass balance approach in conjunction with isotope dilution liquid chromatography electrospray ionization mass spectrometry for accurate quantification. The results showed that the fate of TCS during AS sewage treatment is mainly partitioned into three parts [43]. First, about half of the TCS was adsorbed and accumulated in AS (50 ± 19%); second, about 48 ± 19% of the TCS was actually transformed, lost or unaccounted for during treatment; third, an additional 2 ± 1% of the TCS was discharged into local surface water with the effluent [43]. Thus, it is difficult to achieve the complete removal of TCS by WWTPs.

TCS enters the environment through the effluent of WWTPs. It has been frequently detected in rivers, oceans and other surface water, although TCS has good lipophilic properties and is easily absorbed into sediment and sewage sludge (Appendix A and Table 2). Previous investigations have shown that TCS has a high detection rate in the environment [44]. A large survey conducted by the Joint Research Centre in 10 countries at 686 sampling sites found that TCS was detected in over 40% inland river samples [45]. In the US, the TCS detection rate in 139 rivers was 57.6% [46]. In China, the detection rate of TCS in the Yellow River, Yangtze River, Pearl River, Bohai Sea and East China Sea is more than 90% [45,46,47,48,49,50,51].

Currently, TCS has also been found to accumulate in algae, fishes, birds and other organisms [51,66,67]. Pashaei et al. [28] found that TCS has a concentration of 50–145 μg/kg in aquatic organisms such as Atlantic salmon, Atlantic sea wolf, rainbow trout, Atlantic cod, White vannamei prawn, Indian prawn and kiddi shrimp by HPLC-MS. TCS has also been detected in agricultural ecosystems with different trophic levels following the application of the WWTP biosolids to agricultural field, which means that TCS could be transferred through the terrestrial food web [31,68,69]. TCS was detected in human urine samples in China, the US, India, South Korea and other countries, with the highest average concentration of TCS in urine in China (100 ng/mL), while the lowest was in Vietnam (2.34 ng/mL) [70].

According to the published survey data, TCS is an emerging pollutant with a high frequency detected in the environment. Low concentrations of TCS can affect microorganisms and fish in aquatic environments and can be transmitted to humans through the food web, resulting in an adverse effect on human life and health. Therefore, it is necessary to study the degradation of TCS, especially its biodegradation.

## 3. TCS Biodegradation by Microorganisms

Microbial degradation of pollutants in wastewater systems is the main approach for wastewater environmental bioremediation [71,72]. The degradation of TCS in the aquatic environment by microorganisms is the major method. TCS-degrading microorganisms could be enriched and domesticated from TCS-contaminated water, and TCS can be degraded either by single bacteria or microbial consortium. In this part, the phylogenetic diversity and degradation characteristics of TCS-degrading isolates, diversity and functions of TCS-degrading microbial consortia, as well as the metabolic pathways and mechanisms of TCS-degrading microorganisms have been comprehensively summarized.

### 3.1. TCS Biodegradation by Isolates and Microbial Consortia

#### 3.1.1. Species and Degradation Characteristics of TCS-Degrading Microbial Isolates

Since 2001, the first TCS-degrading bacterium *Sphingomonas* sp. RD1 was isolated from AS of WWTP by Hay et al. [73]. So far, a total of 19 strains of microorganisms isolated from different habitats such as AS, wastewater and soil have been reported with TCS-degrading abilities. They are mainly *Sphingomonas*, *Pseudomonas*, and *Rhodococcus*, etc. (Figure 2, Appendix A).

From the perspective of phylogenetic diversity, there are 15 genera of TCS-degrading microorganisms reported. Among the 15 genera represented, *Pseudomonas* and *Sphingomonas* account for a large proportion of isolates (16.67% each). In terms of degradation characteristics, TCS-degrading bacteria reported so far degrade TCS under aerobic condition except for *Shewanella putrefaciens* CN32 [75]. However, different TCS-degrading microorganisms have different degradation abilities. For example, *Sphingopyxis* sp. KCY1 isolated by Lee et al. [81] from the AS of WWTP was the first reported microorganism which could completely metabolize TCS in wastewater. Strain KCY1 could degrade about 90% of TCS at the initial concentration of 5 mg/L within 24 h and achieve about 100% removal rate on the second day, and could completely dechlorinate TCS [81]. However, *Pseudomonas aeruginosa* KS2002, isolated from AS by Kumari et al. [76], was able to transform 99.89% of TCS into 2,4-dichlorophenol within 96 h when the initial concentration was 2 g/L, but the strain KS2002 could not completely mineralize TCS [76].

The cometabolism phenomenon widely exists in the degradation process of organic pollutants. Currently, it has been reported that microorganisms that can co-metabolically degrade TCS mainly include *Sphingomonas* [82], *Rhodococcus* [83], *Mycobacterium* [83] and so on. The degradation of TCS by these strains was only induced with the presence of primary substrates, such as propane and diphenyl ether. For example, *Sphingomonas* sp. PH-07 could not utilize TCS as the sole carbon source, but TCS could be transformed to fewer toxic compounds by the strain PH-07 when its activity was induced by diphenyl ether [82]. In addition, previous studies have shown that AOB could degrade TCS in the wastewater [1,80,89]. For example, *Nitrosomonas europaea* ATCC 19178, an AOB isolated from a WWTP system, could degrade about 70% of TCS [1].

At present, TCS cannot be completely mineralized by part of degrading bacteria, and TCS cannot be utilized as the sole carbon source and energy for their growth. Therefore, it is necessary to continue to screen more efficient TCS-degrading microorganisms, enrich the resources of TCS-degrading microbial strains, and to improve their application potential in the environmental bioremediation.

#### 3.1.2. TCS Biodegradation by Microbial Consortia

The complete mineralization of TCS requires different metabolic enzymes, and some individual microorganisms cannot accomplish this process independently without the cooperation of the others. But in the actual WWTPs, bacteria often exist in the form of consortia. TCS-degrading bacteria and non-degrading bacteria can completely degrade pollutants through synergistic interaction [90,91]. At present, TCS-degrading enrichment cultures are mainly enriched from environmental samples with TCS as the sole carbon source. The characteristics of TCS-degrading microbial consortia and their analysis approaches are summarized in Table 3.

With the development of high-throughput sequencing, 16S rRNA gene amplicon and metaomics sequencing, this has enabled us to analyze the diversity and community composition of native bacterial communities in TCS polluted water environments. Hay et al. [73] used TCS as substrate to enrich and domesticate a TCS-degrading bacterial consortia from the AS of Neyland WWTPs, and analyzed the community composition of a TCS-degrading bacterial community by using fingerprints such as BOX-PCR and RISA. In addition to metaomics, DNA-SIP can identify functional bacteria in complex mixed systems. Dai et al. [101] used DNA-SIP to identify TCS-degrading bacteria in nitrifying SBR and a lab-scale A/O system, and found that *Amaricoccus* and *Methylobacillus* were the main TCS-degrading bacteria, respectively. Right after, they applied DNA-SIP combined with oligonucleotide typing technology to conduct batch experiments on the TCS biodegradation mechanism, and found that TCS was mainly removed by metabolism of heterotrophic bacteria (accounting for about 62%) [101].

At present, most studies on the TCS-degrading microbial community have been carried out by DNA-SIP, metagenomic and other omics methods to directly study the microbial community structure and community diversity in the process of domestication. However, the specific function of single TCS-degrading microorganism and its interaction with other non-degrading microorganisms in community has not been characterized.

### 3.2. Mechanisms of TCS-Degrading Microorganisms

#### 3.2.1. Metabolic Pathways of TCS-Degrading Microorganisms

The biodegradation pathway of TCS has only been studied in few strains (Figure 3). TCS microbial metabolic pathways that have been proposed include the *meta*-cleavage pathway (Figure 3A), *ortho*-cleavage pathway (Figure 3B), and other unclassified degradation pathways (Figure 3C).

In the *meta*-cleavage pathway, the initial attack of a regioselective dioxygenase at the 2,3-position of TCS results in the formation of dihydroxy-TCS [76,81]. Dihydroxy-TCS can be further transformed to monohydroxy-TCS [76,81]. Pfeifer et al. [102] reported that a further dioxygenation with simultaneous ether-bond cleavage occurred during diphenyl ether degradation. Therefore, monohydroxy-TCS and dihydroxy-TCS may be further linked by 2,3-dioxygenase, followed by ether cracking to produce 2,4-dichlorophenol. 2,4-Dichlorophenol is then converted into small carboxylic acid through ring-opening and is dechlorinated to enter the TCA cycle so that these degrading microorganisms can achieve the mineralization of TCS. During the TCS biodegradation by strain *Sphingopyxis* sp. KCY1 isolated by Lee et al. [81], 6-chloro-3-(2,4-dichlorophenoxy)-4-hydroxycyclohexa-3,5-diene-1,2-dione, monohydroxy-TCS, 2,4-dichlorophenol and other intermediate metabolites were identified by GC/MS. Additionally, four metabolites, monohydroxy-TCS, dihydroxy-TCS, 2,4-dichlorophenol and 2-chlorohydroquinone, were observed during TCS degradation by *Rhodococcus jostii* RHA1 co-cultured with biphenyl as a carbon source [82]. Using mass spectroscopy and Fourier Transform Infrared Spectroscopy, Kumari et al. [76] discovered that *Pseudomonas aeruginosa* KS2002 could convert TCS into 2,4-dichlorophenol. In addition, there are few studies on the degradation pathway of TCS by the ortho-cleavage pathway. Kumari et al. [74] only detected catechol-1,2-dioxygenase activity and analyzed the degradation products of *Citrobacter freundii* KS2003. It is speculated that TCS might be converted into 2,4-dichlorophenol and 3-chloro-*cis*,*cis*-muconate through *ortho*-cleavage pathway, and 2,4-dichlorophenol is then transformed into phenol and catechol through ring-opening process and dechlorination [74].

In addition to the *meta*-cleavage pathway and *ortho*-cleavage pathway, some TCS-degrading bacteria do not follow the above two pathways to degrade TCS. The bacterium *Dyella* sp.WW1 can introduce hydroxyl radicals into the benzene ring during the TCS degradation process, which might be catalyzed by monooxygenase or dioxygenase [79]. Besides, the detection of the release of chlorine ions by ion chromatography and the detection of intermediate metabolites indicates that dechlorination occurs during TCS degradation, but the mechanisms involved are not clear (Figure 3C pathway a) [79]. In addition, in 2021, Balakrishnan and Mohan [18] used LC-MS/MS to detect the intermediates of TCS degradation by *Providencia rettgeri* MB-IIT, and found that methylation (+CH_3_) and complete mineralization were the two possible degradation pathways of the strain MB-IIT. The hydroxyl of TCS was initially biomethylated, and then transformed into downstream intermediates through combined dechlorination [18]. The laccase (LAC) of strain MB-IIT is responsible for transferring molecular oxygen to open up the C-C bond in TCS [18]. The intermediate metabolites produced by the ether bond cleavage and dechlorination are further degraded into a low molecular carboxylic acid and mineralized (Figure 3C pathway b) [18].

Thus, TCS degradation by microorganisms is mainly through oxygenation to open the aromatic ring, and through dechlorination to carry out the biotransformation of toxic intermediates. However, the metabolic pathways of different TCS-degrading strains might be different. In the follow-up analysis of the metabolic pathway of the new isolated TCS-degrading microorganisms, LC-MS/MS and other methods should be performed to analyze the metabolic intermediates and monitor the products’ formation, so as to predict the pathways of TCS degradation by the single isolate.

#### 3.2.2. Enzymes Responsible for TCS Degradation

Enzymes involved in the degradation of TCS are divided into specific enzymes and non-specific enzymes. The only specific enzyme is TCS oxygenase (TcsA). Non-specific enzymes include dioxygenases, monooxygenases, dechlorinating enzymes, and others. The researchers used different approaches to identify the enzymes involved in the TCS degradation, and then studied the biodegradation mechanism of TCS (Table 4).

##### Specific Enzyme Responsible for TCS Biodegradation

As for the studies of specific functional enzymes, a novel TCS oxygenase TcsA was the only biochemically identified protein responsible for TCS biodegradation [103]. Gene *tcsA* encoding the putative TCS oxygenase large subunit was identified in a TCS-degrading fosmid clone from the DNA library of *Sphingomonas* sp. RD1 [103]. Hydroxy-TCS and chlorocatechols were produced during TCS biodegradation by strain RD1, and TcsA was induced by TCS [103]. BLASTp analysis revealed that the best match to TcsA was that of the Rieske [2Fe-2S] cluster domain-containing protein from *Sphingopyxis* sp. MC1, with 100.00% amino acid identity. *Sphingopyxis* sp. MC1 has been annotated as a TCS-degrading bacterium, but its TCS-degrading characteristics have not been described in the literature except for the reported full genome sequence of strain MC1. The Rieske-type aromatic dioxygenase is a multi-component enzyme system with a Rieske [2Fe-2S] center, mostly composed of a terminal oxidase, reductase and ferredoxin [105]. Rieske type aromatic dioxygenases are mainly divided into five categories, including two-component IA, two-component IB, three-component IIA, three-component IIB as well as three-component III [105,106]. Figure 4 lists the evolutionary relationship between TcsA and the representative sequences of Rieske-type aromatic dioxygenases in five groups. The basic information of the reference sequences is shown in Appendix A. According to the phylogenetic relationship, TcsA of *Sphingomonas* sp. RD1 belongs to the Rieske-type two-component class IA aromatic dioxygenase (Figure 4).

##### Non-Specific Enzyme Responsible for TCS Biodegradation

(1)Dioxygenase

Dioxygenases catalyze the incorporation of both oxygen atoms into substrate. According to the catalytic properties, they can be divided into catechol dioxygenases, dihydroxylated aromatic dioxygenases, α-ketoglutarate-dependent dioxygenases, etc. Dioxygenase is involved in the aromatic ring-opening process of TCS biodegradation. Among them, catechol 2,3-dioxygenase (C23O) is involved in the *meta*-cleavage pathway of TCS. During TCS degradation, it was found that only the activity of C23O, not catechol 1,2-dioxygenase (C12O), was detected in the cell extract of *Sphingopyxis* sp. KCY1 (specific enzyme activity = 337 nmol min^−1^ mg protein^−1^) [81]. Besides, the degradation of TCS ceased after the addition of 3-flurocatechol (an inhibitor of C23O that catalyzes *meta*-cleavage reactions) at 23 h [81]. Kumari et al. [76] detected C23O (specific enzyme activity = 0.161 U mg^−1^) in the cell-free extract of *Pseudomonas aeruginosa* KS2002, and TCS degradation was also ceased in the presence of 3-flurocatechol. These results indicate that C23O is involved in TCS biodegradation via *meta*-cleavage pathway.

C12O is involved in the *ortho*-cleavage pathway of TCS. Kumari et al. [74] detected the activity of C12O (specific enzyme activity =0.159 U mg^−1^) in the cell-free extract of *Citrobacter freundii* KS2003. In addition, TCS could not be degraded in the presence of 4-chlorcatechol, an inhibitor of C12O, suggesting that C12O was involved in the degradation of TCS via *ortho*-cleavage pathway [74].

(2)Monooxygenase

In addition to dioxygenase, some monooxygenases also participate in the TCS biodegradation. Lee et al. [86] found that propane monooxygenase (PMO) of *Mycobacterium vaccae* JOB5 and alkane monooxygenase (AlkMO) of *Rhodococcus jostii* RHA1 might be involved in the TCS degradation. Ammonia monooxygenase (AMO) can also degrade TCS [1]. Roh et al. [1] examined the biodegradation potential of TCS by *Nitrosomonas europaea* and mixed AOB in nitrifying AS. The degradation was observed only in the absence of allylthiourea (an inhibitor for AMO), suggesting that AMO might be responsible for TCS biodegradation [1].

(3)Reductive dehalogenase

At present, only Zhao et al. have studied the dechlorination mechanism of TCS [104]. *Dehalococcoides mccartyi* CG1 could metabolically dechlorinate TCS to diclosan in the AS of the WWTP [104]. Both the tolerance of strain CG1 to TCS and the rate of TCS dechlorination increased when CG1 was cultured in the presence of TCS and tetrachloroethene [104]. A dechlorase PcbA1 was identified by nano-LC-MS-MS and an in vitro activity assay, which was found to be responsible for the TCS dechlorination in strain CG1 [104].

(4)Others

There are also other enzymes involved in the biodegradation of TCS. Balakrishnan et al. [18] reported that the enzyme activities of MnP and LAC were detected during the TCS degradation by *Providencia rettgeri* MB-IIT, indicating that MnP and LAC might concurrently participate to the TCS degradation.

At present, although many TCS-degrading microorganisms have been isolated, most studies have not been carried out at the molecular level. Subsequently, the enzyme involved in TCS degradation should be studied by molecular biological methods, and its specific function should be verified by gene knockout, complementation, and in vitro enzyme activity, so as to explore the molecular mechanisms of TCS biodegradation.

## 4. Summary and Prospect

TCS, as an emerging pollutant in the water environment, has attracted significant attention since the outbreak of the COVID-19 pandemic [14,25,26,27]. With regard to the removal of TCS in aquatic environments, the microbial-mediated degradation of TCS has been consistently considered as the main method of TCS bioremediation. In view of the problems and limitations existing in the current research, some prospects are proposed: (1) Although many TCS-degrading strains have been isolated and characterized from different environments, the metabolic pathways and detailed molecular mechanisms of TCS biodegradation, especially the specific enzymes and functional genes, have not been fully and deeply studied. Therefore, it is still necessary to isolate and screen much more efficient TCS-degrading microorganisms and to characterize their degradation characterizations through molecular biological methods, including the identification of new enzymes, exploration of the catalytic mechanisms and modification of enzymes, etc., to elucidate the molecular regulation mechanism of TCS degradation, and further improve the environmental adaptability, stability, as well as degradation efficiency of the enzymes. (2) Most of the reported bio-augmentation applications have been carried out under laboratory conditions, and no studies have been published on the use of TCS-degrading bacteria in the actual contaminated environments. Therefore, as a promising bioremediation technology, the TCS bio-augmentation strategy needs to be further improved, especially to study the long-term degradation effect of TCS-degrading bacteria inoculation in polluted environments and the effect of bioaugmentation on other organisms in the environment. (3) Synthetic consortium is one of the important issues of microbial degradation of TCS. The application of modern molecular biology techniques such as DNA-SIP and metagenomics provides a favorable method for the exploration of functional TCS-degrading bacteria. With the development of synthetic biology and metabolic modeling, synthetic microbial ecology will allow us to construct artificial TCS-degrading microbial consortia by an iterative Design–Build–Test–Learn approach in the future.

By studying the interaction mechanisms between functional strains, TCS-degrading synthetic bacterial consortia can be constructed by a “Bottom-up” approach, thus providing the robustness of TCS degradation system and alleviating the stress pressure of environmental changes on TCS-degrading microorganisms.

## Figures and Tables

**Figure 1 microorganisms-10-01713-f001:**
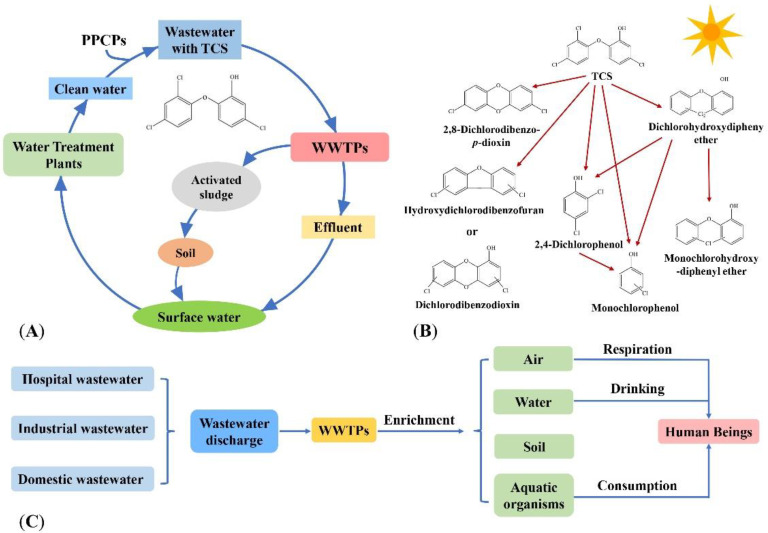
Occurrence of TCS in the water environment. (**A**) The circulation of TCS in the water environment. (**B**) Photodegradation products of TCS [32,33]. Reprinted with permission from Ref. [32]. 2022, Dar et al.; and Ref. [33]. 2008, Sanchez-Prado et al. (**C**) The fate of TCS in the water environment.

**Figure 2 microorganisms-10-01713-f002:**
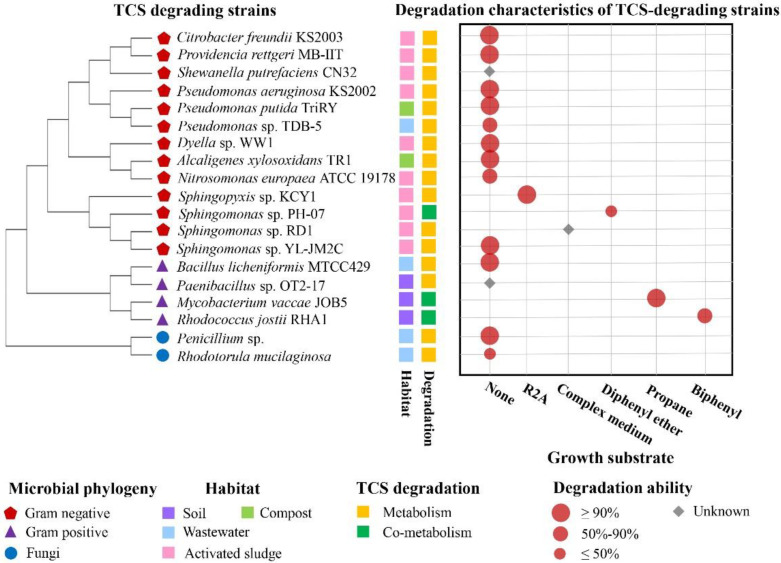
Phylogenetic analyses and degradation characteristics of TCS-degrading microorganisms [1,18,74,75,76,77,78,79,80,81,82,83,84,85,86,87,88]. The phylogenetic tree was built based on 16S rRNA gene sequences covering 19 TCS-degrading microorganisms. Detailed information on the degradation characteristics of typical TCS-degrading isolates is included in Appendix A.

**Figure 3 microorganisms-10-01713-f003:**
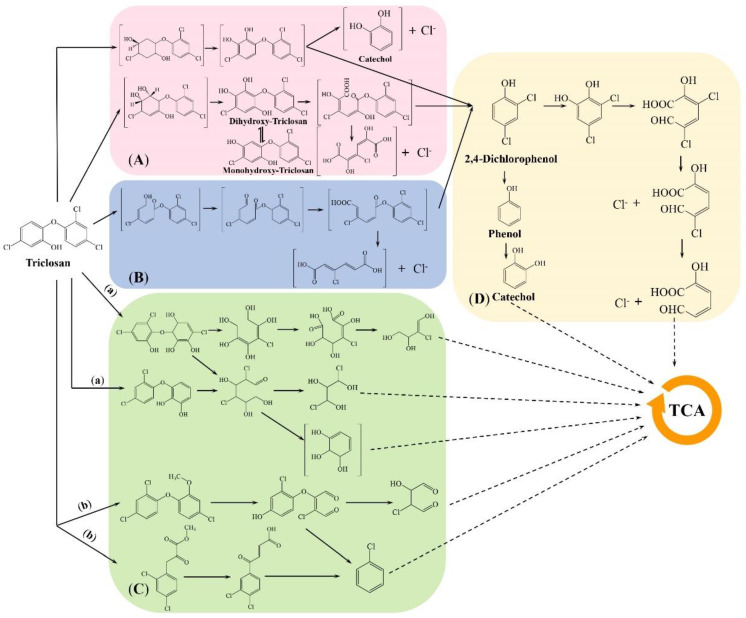
Proposed degradation pathway of TCS by microorganisms. (**A**). *Meta*-cleavage pathways [76,81,83]; (**B**). *Ortho*-cleavage pathways [74]; (**C**). Others: (a). Proposed degradation pathway of TCS by *Dyella* sp. WW1 [79]; (b). Proposed degradation pathway of TCS by *Providencia rettgeri* MB-IIT [18]. (**D**). Common downstream pathway of *meta*-cleavage and *ortho*-cleavage pathways.

**Figure 4 microorganisms-10-01713-f004:**
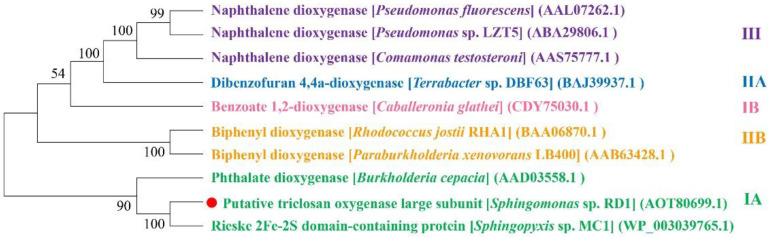
Unrooted phylogenetic tree generated based on amino acid sequences of the large subunit of putative TCS oxygenase [103]. Rieske-type aromatic dioxygenases divided into the same group are represented by the same color. The reference sequences are shown in [107,108,109,110]. The putative TCS oxygenase large subunit analyses of strain *Sphingomonas* sp. RD1 are marked with a red circle.

**Table 1 microorganisms-10-01713-t001:** The concentrations of TCS in WWTPs.

Country	Name of the WWTP	Processing Technology	Concentration of TCS in Wastewater			Concentration of TCS in Sludge	Reference
			Influent	Treated Effluent	Removal Rate/%		
China	Northern China WWTP	Anoxic-aerobic (A/O)	295 ± 4.2 ng/L	39 ± 2.7 ng/L	86.77	1801 ng/g	[38]
Brazil	WWTP A	Activated sludge (AS)	1.30 ± 0.22 μg/L	0.55 ± 0.02 μg/L	57.69%	0.94 μg/L	[39]
	WWTP B	Upflow anaerobic sludge blanket	1.26 ± 0.09 μg/L	0.78 ± 0.05 μg/L	38.10%	2.79 μg/L	
	WWTP C	Waste stabilization pond	1.42 ± 0.04 μg/L	0.39 ± 0.02 μg/L	72.54%	0.53 μg/L	
Chile	WWTP	AS and a pilot plant of horizontal subsurface flow	0.20 ± 0.06 μg/L	0.02 ± 0.01 μg/L	90.00%	0.01 ± 0.01 μg/L	[40]
India	WWTP 1	AS	N.A. ^1^	N.A. ^1^	39–62%	N.A. ^1^	[41]
	WWTP 2				45–55%		
China	WWTP#1	A/O	59–1100 ng/L Mean 274 ng/L	13–110 ng/L Mean 83 ng/L	69.71%	N.A. ^1^	[42]
	WWTP#2	Hydrolytic acidification and cyclic activated sludge technology	230–2900 ng/L Mean 389 ng/L	9–180 ng/L Mean 17 ng/L	95.63%		

^1^ ‘N.A.’ means that the relevant data are not available.

**Table 2 microorganisms-10-01713-t002:** The concentrations of TCS in different water environment.

Environment	Method	TCS Concentration	Country	Year of the TCS Determination	Reference
River water	Liquid chromatography-tandem mass spectrometry (LC-MS/MS)	N.D. ^1^ −62.124 µg/L	India	2019–2020	[52]
River water	LC-MS/MS	N.D. ^1^ −135 ng/L Mean 25.4 ng/L	China	2018, 2019 and 2021	[49]
River water	Gas chromatography-mass spectrometer (GC-MS)	0.06–500 ng/L Mean 176.2 ng/L	Morocco	2019	[53]
River water	LC-MS/MS	Up to 74.3 µg/L	India	/ ^2^	[54]
River water	High-performance liquid chromatography-tandem mass spectrometry (HPLC-MS/MS)	N.D. ^1^ −1761 ng/L Mean 942 ng/L in monsoon season	India	2018–2019	[55]
River water	Literature data collection	N.D. ^1^ −293.64 ng/L	China	2010–2019	[56]
River water	LC-MS/MS	0.69–17.5 ng/L	China	2019	[48]
River water	LC-MS/MS	5.1–874 ng/L Mean 0.06 nM	Canada	2012–2013	[57]
River water	HPLC-MS/MS	N.D. −65.6 ng/L Mean 0.02 nM	China	2015	[58]
River water	LC-LC-MS/MS	N.D. ^1^ −0.77 nM	Spain	2012	[59]
River water	High performance liquid chromatography with photo diode array detection	N.D. ^1^ −3.87 nM	South Africa	/ ^2^	[60]
River water	HPLC-MS/MS	0.349 ± 0.032 nM	UK	/ ^2^	[61]
River water	GC-MS	0.01–0.207 nM	Denmark	2010	[62]
Sea water	LC-MS/MS	N.D. ^1^ −58.3 ng/L Mean 22.3 ng/L	China	2018, 2019 and 2021	[49]
Sea water	Ultra-performance liquid chromatography coupled to a triple quadrupole mass spectrometry	N.D. ^1^ −8.7 ng/L Mean 4.2 ng/L	China	2019	[50]
Underground water	LC-MS/MS	0.5–13.1 ng/L Mean 2.9 µg/L	Poland	2019	[63]
Drinking water	LC-MS	Up to 9.74 ng/L	Malaysia	2018	[64]
Drinking water	GC-MS	0.6–9.7 ng/L	China	/ ^2^	[65]

^1^ ‘N.D.’ means that the relevant data are not detected. ^2^ ‘/’ means that the relevant information is not mentioned.

**Table 3 microorganisms-10-01713-t003:** Composition of TCS-degrading microbial communities and research methods.

Environmental Matrices	TCS Initial Concentration	Potential TCS-Degrading Taxa	Community Analysis Approaches	Reference
Bioreactor	20 μg/g	*Flavobacterium*, *Thermomicrobia*, *Nonomuraea* and *Fluviicola*	16S rRNA amplicon sequencing and metagenomic sequencing	[92]
Nitrifying sequencing batch reactors (SBRs) and denitrifying SBRs	4 mg/L	*Sphingomonadaceae*	DNA stable isotope detection (DNA-SIP), 16S rRNA amplicon sequencing and oligotyping analysis	[93]
SBR	3 mg/L	*Thauera*	DNA-SIP and 16S rRNA amplicon sequencing	[94]
Partial nitrification-anammox process	0.5 mg/L	*Alicycliphilus* and *Sphingopyxis*	DNA-SIP and 16S rRNA amplicon sequencing	[95]
Immobilized cell bioreactor	400 mg/L	*Bradyrhizobiaceae*, *Ferruginibacter*, *Thermomonas*, *Lysobacter* and *Gordonia*	16S rRNA amplicon sequencing	[96]
Sludge	4 mg/L	*Sphingobium*	DNA-SIP, 16S rRNA amplicon sequencing, oligotyping analysis and metagenomic sequencing	[97]
SBR	100 μg/L	*Flavobacterium*	16S rRNA amplicon sequencing	[98]
Sludge and wastewater	5.8 g/L	*Chloroflexi*, *Smithella* and *Pseudomonas*	16S rRNA amplicon sequencing	[99]
SBR	100 μg/L	Unidentified *Xanthomonadaceae*, unidentified *Rhizobiales*, *Lacibacter*, and *Dechloromonas*	16S rRNA amplicon sequencing and fluorescent in situ hybridization	[100]
A lab-scale A/O System	2, 4, 6 and 8 mg/L	*Candidatus* Microthrix, *Flavobacterium*, *Thiothrix*	DNA-SIP and 16S rRNA amplicon sequencing	[101]
AS	500 mg/L	*Sphingomonas*	Genomic fingerprinting, Ribosomal intergenic spacer analysis (RISA)	[73]

**Table 4 microorganisms-10-01713-t004:** Methods to identify the different enzymes involved in TCS biodegradation.

Classification		Enzyme	Method of Identification	Strain	Catabolic Pathways	Reference
Specific enzyme	Oxygenase	TCS oxygenase (TcsA)	Constructing TCS-degrading fosmid clone	*Sphingomonas* sp. RD1	Converting TCS into 2,4-dichlorophenol and 4-chlorocatechol	[103]
Non-specific enzyme	Dioxygenase	Catechol 2,3-dioxygenase (C23O)	Adding inhibitors of C23O; detecting enzyme activity	*Sphingopyxis* sp. KCY1	*Meta*-cleavage pathway	[81]
			Adding inhibitors of C23O; detecting enzyme activity	*Pseudomonas aeruginosa* KS2002		[76]
		C12O	Adding inhibitors of C12O; detecting enzyme activity	*Citrobacter freundii* KS2003	*Ortho*-cleavage pathway	[74]
	Monooxygenase	Propane monooxygenase (PMO)	Adding inhibitors of PMO	*Mycobacterium vaccae* JOB5	/ ^1^	[86]
		Alkane monooxygenase (AlkMO)	Adding inhibitors of AlkMO	*Rhodococcus jostii* RHA1	/ ^1^	
		AMO	Adding inhibitors of AMO	*Nitrosomonas europaea* ATCC 19178	/ ^1^	[1]
	Peroxidase	Manganese peroxidase (MnP)	Detecting enzyme activity	*Providencia rettgeri* MB-IIT	Ether bond cleavage	[18]
	LAC	LAC	Detecting enzyme activity		Transferring molecular oxygen	
	Reductive dehalogenase	Reductive dehalogenase (PcbA1)	Proteome; detecting enzyme activity in vitro	*Dehalococcoides mccartyi* CG1	Dechlorination (convert TCS into diclosan)	[104]

^1^ ‘/’ means that the relevant information is not mentioned.

## Data Availability

Not applicable.

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
