# Peer review of "Degradation of Triclosan in the Water Environment by Microorganisms: A Review"

_microorganisms, 2022, doi:10.3390/microorganisms10091713_

Round 1

Reviewer 1 Report

My specific comments to the authors are:

1)Please do not include more than 6-7 sentences in one paragraph. It makes reading difficult for the readers.

2) Please improve Fig.1. It looks amateurish. Incorporate more information dedicated to TCS.

3)In section, 4, more body(text) should be added to the section instead of the table itself. Try to incorporate more recent references.

4)A meta-analyses study of the discussed results on TCS degradation over the last decade would add depth to the review. 

Author Response

Reviewer 1:

Comments and Suggestions for Authors:

My specific comments to the authors are:

1) Please do not include more than 6-7 sentences in one paragraph. It makes reading difficult for the readers.

Response: Thank you very much for your comments. We have rewritten each paragraph including no more than 6-7 sentences.

2) Please improve Fig.1. It looks amateurish. Incorporate more information dedicated to TCS.

Response: Thank you very much for this constructive suggestion. We have redrawn Figure 1 and divided Figure 1 into three parts: “The circulation of TCS in the water environment”, “Photodegradation products of TCS” and “The fate of TCS in the water environment”. The first part of Figure 1 depicts the cycle of TCS in the water environment, aiming to illustrate the occurrence of TCS. The second part of Figure 1 depicts the photolysis products of TCS and intends to illustrate that TCS in the water environment could be converted into toxic intermediates that are harmful for organisms under light conditions. The third part of Figure 1 describes the fate of TCS and aims to illustrate how TCS in the environment is harmful for human beings.

3) In section 4, more body(text) should be added to the section instead of the table itself. Try to incorporate more recent references.

Response: We thank the Reviewer for this constructive suggestion. We have integrated TCS biodegradation by microbial consortia with section 3.1, since relevant studies of TCS-degrading microbial consortia are insufficient. Therefore, we have revised the title of 3.1 to “TCS biodegradation by isolates and microbial consortia”, and divided 3.1 into two parts, namely “3.1.1. Species and degradation characteristics of TCS-degrading microbial isolates” and “3.1.2. TCS biodegradation by microbial consortia”.

4) A meta-analyses study of the discussed results on TCS degradation over the last decade would add depth to the review.

Response: According to the requirements of the Microorganisms, reviews using meta-analysis are systematic reviews. Systematic reviews should follow the PRISMA guidelines. Our paper is a general review, not systematic review with meta-analysis. Therefore, we collected the literatures related to TCS biodegradation, and did a keyword co-occurrence network analysis of the 45 references related to TCS biodegradation involved in this review to analyze the keywords with high frequency. The result is shown in Lines 76-87, Figure S1 and Table S2, respectively.

Reviewer 2 Report

This manuscript reports on work highly relevant to the field of Microorganisms. The review is interesting, up-to-date, and easy to follow; however, I would recommend that the manuscript be accepted after major revisions because some details still require more careful explanation.

My specific comments:

1. Figure 1 and Table 1 have almost the same title. It should be differentiated.

2. Title of Fig.1 is not giving information about what is on it. It presented the sources of TCS in water, the food chain, and human beings. I would suggest rephrasing to be more specific.

3. Table 2. The title should be more precise. Some references are not provided (Error! Reference source not found). Year - it is unclear if it is the year of publication or the year of the TCS determination. Please clarify.

4. Figure 2. It is not clear the source of the data presented, references are missing.

5. Figure 3. should be placed just after the first mention in the text, just after line 172. Now it is not easy to follow.

6. Fig. 3 and Fig. 4. should have cited references, the basis on which have been prepared.

7. Table 3 References are not provided (some errors occur after formatting the text)

8. Supplementary materials. Authors should be consistent with units, In the main body of the manuscript, there is mg/L, in Supplementary materials mg L-1

Author Response

Reviewer 2:

Comments and Suggestions for Authors:

This manuscript reports on work highly relevant to the field of Microorganisms. The review is interesting, up-to-date, and easy to follow; however, I would recommend that the manuscript be accepted after major revisions because some details still require more careful explanation.

Response: Thank you for reviewing this manuscript. We appreciate your input as peer review always improves the quality of manuscript. We have carefully considered each point and have made efforts to answer the questions.

My specific comments:

  1. Figure 1 and Table 1 have almost the same title. It should be differentiated.

Response: We thank the Reviewer for this valuable suggestion. We have redrawn Figure 1 and changed the title to “Figure 1. Occurrence of TCS in the water environment. (A) The circulation of TCS in the water environment. (B) Photodegradation products of TCS [32,33]. (C) The fate of TCS in the water environment.”, see Lines 175-178 for details. Besides, we have revised the title of Table 2 (previously Table 1) to “Table 2. The concentrations of TCS in different water environment”.

  1. Title of Fig.1 is not giving information about what is on it. It presented the sources of TCS in water, the food chain, and human beings. I would suggest rephrasing to be more specific.

Response: We thank the Reviewer for this question. As suggested, we have rewritten the title of Figure 1 and divided Figure 1 into three parts: “The circulation of TCS in the water environment”, “Photodegradation products of TCS” and “The fate of TCS in the water environment”. The first part of Figure 1 depicts the cycle of TCS in the water environment, aiming to illustrate the occurrence of TCS. The second part of Figure 1 depicts the photolysis products of TCS and intends to illustrate that TCS in the water environment could break down into intermediates that are toxic to organisms under light conditions. The third part of Figure 1 describes the fate of TCS and aims to illustrate how TCS in the environment is harmful for humans.

  1. Table 2. The title should be more precise. Some references are not provided (Error! Reference source not found). Year - it is unclear if it is the year of publication or the year of the TCS determination. Please clarify.

Response: Table 2 (previously Table 1), where “Year” refers to the time to detect TCS, has been corrected in the table. And we have added the missing references.

  1. Figure 2. It is not clear the source of the data presented, references are missing.

Response: The specific information of the TCS-degrading bacteria is shown in Table S2, and the references has been cited in the title of Figure 2.

  1. Figure 3. should be placed just after the first mention in the text, just after line 172. Now it is not easy to follow.

Response: We thank the Reviewer for this question. We have placed Figure 3 after Line 817.

  1. Fig. 3 and Fig. 4. should have cited references, the basis on which have been prepared.

Response: We agree with the Reviewer for this question. We add the references in the title of Figures 3 and 4, with specific information in Lines 817-822 and Lines 1066-1069, respectively.

  1. Table 3 References are not provided (some errors occur after formatting the text).

Response: We thank the Reviewer for this question. We have added the missing references in Table 3.

  1. Supplementary materials. Authors should be consistent with units, In the main body of the manuscript, there is mg/L, in Supplementary materials mg L-1.

Response: We thank the reviewer for this question. We have unified the units involved in the supplementary materials.

Reviewer 3 Report

The manuscript “degradation of triclosan in the water environment by microorganisms: a review” has reviewed the occurrence, microorganisms capable of TCS degradation, the pathways and enzymes involved in the biodegradation. As the more intensive use of triclosan as disinfectants in the COVID-19 pandemic, it is scientifically valuable and of great public concern to have a review now. The current version provided some helpful summary about microorganisms, pathways and mechanisms in TCS degradation. However, more information is expected from a review, and some information is not concisely organized. The authors should carefully address the issues before the further publication of this manuscript.

Major comments:

The impact why we should care about TCS biodegradation should be discussed. The information, such as the adverse effects of TCS on both humans and ecosystem and the environmental-relevant toxic concentrations, should be at least introduced in introduction. 

Section 2 “the occurrence of TCS in water environment”, the role of WWTPs which is the primary TCS treatment system is not fully discussed. The reviewer would suggest adding a table which summarizes the TCS concentration in influent and effluent. It would be even better to slightly discuss the TCS adsorption to the sludge to state the full route of TCS in WWTPs

Section 4 “TCS biodegradation by microbial consortia”. Since the authors list TCS degradation by isolates and microbial consortia as two sections, the interaction among microbial consortia is actually expected. However, such information was not fully discussed. If relevant studies of the microbial interactions is insufficient , the reviewer would suggest the authors integrating TCS biodegradation by microbial consortia with section 3.1. 

Specific comments:

Lines 49-50. Unprecise description. Nitrogen removal bacteria and heterotrophic bacteria are categories by different classifications.

Lines 69-74. Unprecise description. this is a process called bioaccumulation. 

Lines 83-84. Please provide citation to support this statement.

Lines 83-86. please provide physical characteristics for a clearer comparison of the water/solvent solubility.

Table 1. the minimum and maximum concentrations of TCS should also be provided if available.

Figure 2. this figure summarized the microorganisms which are reported to degrade TCS. The reviewer would suggest the authors to include a table adjacent to the described strains to better present their degrading characteristics instead of the current version. Besides gram-positive or negative, habitat, metabolism/co-metabolism, and degradation percentage which are already there, whether it is an aerobic or anaerobic process and degrading end products should also be included. Please note that the degradation percentage is only meaningful when the initial dosage concentration is provided. 

Lines 111-112. The connection between the statement of “therefore, biodegradation of TCS has been considered as an environmentally friendly” and the sentences above is not logical.

The description of TCS degrading strains is unclear. Only Pseudomonas and Sphingomonas are discussed. None of the other microorganisms which metabolically degrade TCS are discussed, or even described. 

Lines 149-152: Unclear description. How strains can be induced? Are the authors trying to say “the degradation of TCS by these strains was only induced with the presence of primary substrates, such as propane and diphenyl ether”

Lines 155-160. This describes degradation mechanisms and does not belong here.

Lines 161-164. This statement is purely speculation and lacks logical connections. 

Section 3.2: “The mechanisms of TCS-degrading microorganisms” should be separated as an individual section. The enzymes may be introduced by different categories.

Lines 252-254: Actually, there are different approaches to identify an enzyme in a process. The reviewer is confused of the authors’ emphasis on “mechanisms at the molecular level” and would suggest the authors to remove this emphasis. Instead, the authors may add a summary table of the enzymes involved in TCS degradation by listing the identification approach as well. 

Figure 4. the reviewer is curious about the phylogeny of other discussed enzymes in TCS degradation.

Line 249. Change “dechlorase” to “reductive dehalogenase”. This reductive dehalogenase in strain CG1 is identified and characterized as the authors say, “molecular level”.

Please check the citations very carefully. There are obvious references missing in tables.

Author Response

Reviewer 3:

Comments and Suggestions for Authors:

The manuscript “degradation of triclosan in the water environment by microorganisms: a review” has reviewed the occurrence, microorganisms capable of TCS degradation, the pathways and enzymes involved in the biodegradation. As the more intensive use of triclosan as disinfectants in the COVID-19 pandemic, it is scientifically valuable and of great public concern to have a review now. The current version provided some helpful summary about microorganisms, pathways and mechanisms in TCS degradation. However, more information is expected from a review, and some information is not concisely organized. The authors should carefully address the issues before the further publication of this manuscript.

Response: Thank you for your constructive suggestions, which are very helpful to improve the quality of our manuscript. As suggested, we have rewritten some unclear descriptions in the manuscript. We have added the role of WWTPs in TCS degradation in Section 2 and added a table illustrating the role of WWTPs. In addition, we merged the original Section 4 and 3.1, and divided 3.1 into two parts to describe the degradation of TCS-degrading bacteria and microbial consortia, respectively. What’s more, we rearranged the paragraphs about TCS-degrading enzymes in 3.2, and rewritten them according to the type of TCS-degrading enzymes.

Major comments:

The impact why we should care about TCS biodegradation should be discussed. The information, such as the adverse effects of TCS on both humans and ecosystem and the environmental-relevant toxic concentrations, should be at least introduced in introduction.

Response: We thank the Reviewer for this question. As suggested, we have added the environmental toxicology study of TCS in Lines 33-38, aiming to explain why we should care about TCS biodegradation.

Section 2 “the occurrence of TCS in water environment”, the role of WWTPs which is the primary TCS treatment system is not fully discussed. The reviewer would suggest adding a table which summarizes the TCS concentration in influent and effluent. It would be even better to slightly discuss the TCS adsorption to the sludge to state the full route of TCS in WWTPs.

Response: We thank the Reviewer for this constructive suggestion. WWTPs played an important role in the degradation of TCS. We have added a table (Table 1) which summarizes the TCS concentration in influent and effluent of WWTPs.

Section 4 “TCS biodegradation by microbial consortia”. Since the authors list TCS degradation by isolates and microbial consortia as two sections, the interaction among microbial consortia is actually expected. However, such information was not fully discussed. If relevant studies of the microbial interactions is insufficient, the reviewer would suggest the authors integrating TCS biodegradation by microbial consortia with section 3.1.

Response: We thank the Reviewer for this constructive suggestion. We have integrated TCS biodegradation by microbial consortia with section 3.1, since relevant studies of TCS-degrading microbial consortia are insufficient. Therefore, we have revised the title to “3.1. TCS biodegradation by isolates and microbial consortia”, and divided 3.1 into two parts, namely “3.1.1. Species and degradation characteristics of TCS-degrading microbial isolates” and “3.1.2. TCS biodegradation by microbial consortia”.

Specific comments:

Lines 49-50. Unprecise description. Nitrogen removal bacteria and heterotrophic bacteria are categories by different classifications.

Response: We are sorry for this unclear description. We checked the original literature (Roh, H.; Subramanya, N.; Zhao, F.M.; Yu, C.P.; Sandt, J.; Chu, K.H. Biodegradation potential of wastewater micropollutants by ammonia-oxidizing bacteria. Chemosphere 2009, 77, 1084-1089) and confirm that nitrogen removal bacteria should be ammonia-oxidizing bacteria. So, we replaced “nitrogen removal bacteria” with “ammonia-oxidizing bacteria”. Given that most ammonia-oxidizing bacteria are autotrophic, thus we have corrected this sentence to “Studies have shown that TCS could be degraded by the ammonia-oxidizing bacteria (AOB) and heterotrophic bacteria in the water environment” in Line 69.

Lines 69-74. Unprecise description. this is a process called bioaccumulation.

Response: Thanks. We have replaced “adsorption” with “bioaccumulation” in Line 111.

Lines 83-84. Please provide citation to support this statement.

Response: We removed the half sentence “About 79% of TCS can be removed by WWTPs,”. And we have rewritten this sentence to “Thus, it is difficult to achieve complete removal of TCS by WWTPs” in Lines 231-232.

Lines 83-86. please provide physical characteristics for a clearer comparison of the water/solvent solubility.

Response: The physical characteristics of TCS have been listed in Table S1. We added the citation of Table S1 in this sentence, which can be shown in Line 235.

Table 1. the minimum and maximum concentrations of TCS should also be provided if available.

Response: Thank you for this attention. We have added the maximum and minimum concentrations of TCS that can be retrieved.

Figure 2. this figure summarized the microorganisms which are reported to degrade TCS. The reviewer would suggest the authors to include a table adjacent to the described strains to better present their degrading characteristics instead of the current version. Besides gram-positive or negative, habitat, metabolism/co-metabolism, and degradation percentage which are already there, whether it is an aerobic or anaerobic process and degrading end products should also be included. Please note that the degradation percentage is only meaningful when the initial dosage concentration is provided.

Response: We thank the reviewer for this valuable comment. The table you mentioned is Table S2, which can be viewed in the Supplementary Materials. Regarding anaerobic or aerobic degradation of TCS, the current reported TCS-degrading microorganisms could degrade TCS under aerobic conditions except for Shewanella putrefaciens CN32 (Line 596). Besides, as suggested, we added “Initial TCS concentration” and “Degrading end products” to Table S2.

Lines 111-112. The connection between the statement of “therefore, biodegradation of TCS has been considered as an environmentally friendly” and the sentences above is not logical.

Response: We thank the reviewer for pointing this out. We have corrected this sentence to “Therefore, it is necessary to study the degradation of TCS, especially the biodegradation.” In Lines 395-396.

The description of TCS degrading strains is unclear. Only Pseudomonas and Sphingomonas are discussed. None of the other microorganisms which metabolically degrade TCS are discussed, or even described.

Response: We are sorry that this section was not clearly described. In the section “3.1.1 Species and degradation characteristics of TCS-degrading microbial isolates”, we described the phylogenetic diversity, degradation abilities and metabolic patterns of the degrading bacteria such as Sphingopyxis sp. KCY1, Pseudomonas aeruginosa KS2002, Sphingomonas sp. PH-07, and Nitrosomonas europaea ATCC 19178, not only Pseudomonas and Sphingomonas. Regarding the specific metabolic pathways and metabolic mechanisms of TCS-degrading bacteria, we will describe in section 3.2.

Lines 149-152: Unclear description. How strains can be induced? Are the authors trying to say “the degradation of TCS by these strains was only induced with the presence of primary substrates, such as propane and diphenyl ether”.

Response: We thank the Reviewer for this valuable comment. It is true that we misrepresented. We have corrected this sentence to “the degradation of TCS by these strains was only induced with the presence of primary substrates, such as propane and diphenyl ether” in Line 609.

Lines 155-160. This describes degradation mechanisms and does not belong here.

Response: We thank the Reviewer for this question. As suggested, we have moved this sentence to Line 1094-1098 and modified it appropriately.

Lines 161-164. This statement is purely speculation and lacks logical connections.

Response: Thank you for this attention. We have corrected this sentence to “At present, TCS cannot be completely mineralized by part of degrading bacteria, and TCS cannot be utilized as the sole carbon source and energy for their growth.” in Lines 615-616.

Section 3.2: “The mechanisms of TCS-degrading microorganisms” should be separated as an individual section. The enzymes may be introduced by different categories.

Response: We thank the Reviewer for this question. As suggested, we have divided “3.2. Mechanisms of TCS-degrading microorganisms” into “3.2.1. Metabolic pathways of TCS-degrading microorganisms” and “3.2.2. Enzymes responsible for TCS degradation”, of which 3.2.2. was further divided into “3.2.2.1. Specific enzyme responsible for TCS biodegradation” and “3.2.2.2. Non-specific enzyme responsible for TCS biodegradation”. The specific enzymes section describes the related research progress of TCS oxygenase, and the non-specific enzymes are divided into “(1) Dioxygenase”, “(2) Monooxygenase”, “(3) Reductive dechlorination” and “(4) Others” according to the type of enzyme.

Lines 252-254: Actually, there are different approaches to identify an enzyme in a process. The reviewer is confused of the authors’ emphasis on “mechanisms at the molecular level” and would suggest the authors to remove this emphasis. Instead, the authors may add a summary table of the enzymes involved in TCS degradation by listing the identification approach as well.

Response: As suggested, we remove “mechanisms at the molecular level” and “molecular mechanisms”. And we have added a table (Table 4) listing the identification approach and catabolic pathways of enzymes for TCS biodegradation.

Figure 4. the reviewer is curious about the phylogeny of other discussed enzymes in TCS degradation.

Response: Sorry for confusion. In addition to the specific enzyme TCS oxygenase (TcsA), other enzymes that can participate in the degradation of TCS such as C12O, C23O, AMO, LAC, dechlorinase, are all non-specific enzymes for TCS degradation. Therefore, we only analyzed the specific enzyme for TCS degradation when performing phylogenetic analysis.

Line 249. Change “dechlorase” to “reductive dehalogenase”. This reductive dehalogenase in strain CG1 is identified and characterized as the authors say, “molecular level”.

Response: Thanks for your comments. We have now replaced “dechlorase” with “reductive dehalogenase” in Lines 1099-1105, thanks.

Please check the citations very carefully. There are obvious references missing in tables.

Response: As suggested, we have now checked and added missing references in Table 2 and 3.

Round 2

Reviewer 2 Report

Thank you for including my remarks, after revision manuscript looks much better. Great job!

Author Response

Reviewer 2:

Comments and Suggestions for Authors:

Thank you for including my remarks, after revision manuscript looks much better. Great job!

Response: Thank you for your positive comments.

Reviewer 3 Report

Most of the concerns are carefully addressed. However, I believe there is content missing in section 3.2.2. "enzyme responsible for TCS degradation", probably due to formatting mistakes? (1) table 4 is missing. (2)five-line contents for a section seems a bit thin and please elaborate the contents. why the description of the enzymes in original version was completely deleted? the origin description is roughly okay.

Author Response

Reviewer 3:

Comments and Suggestions for Authors:

Most of the concerns are carefully addressed. However, I believe there is content missing in section 3.2.2. "enzyme responsible for TCS degradation", probably due to formatting mistakes? (1) table 4 is missing. (2) five-line contents for a section seems a bit thin and please elaborate the contents. why the description of the enzymes in original version was completely deleted? the origin description is roughly okay.

Response: Sorry for the format error caused during processing. We supplemented the missing content in section “3.2.2. Enzyme responsible for TCS degradation”, and uploaded the file without “Track Changes” function, which can be found in Lines 279-357 of the manuscript.